# Evaluation of Bovine Lactoferrin for Prevention of Late-Onset Sepsis in Low-Birth-Weight Infants: A Double-Blind Randomized Controlled Trial

**DOI:** 10.3390/nu17111774

**Published:** 2025-05-23

**Authors:** Shabina Ariff, Sajid Bashir Soofi, Uswa Jiwani, Almas Aamir, Uzair Ansari, Arjumand Rizvi, Michelle D’Almeida, Ashraful Alam, Michael Dibley

**Affiliations:** 1Department of Pediatric and Child Health, Aga Khan University, Karachi 74800, Pakistan; 2Center of Excellence in Women and Child Health, Aga Khan University, Karachi 74800, Pakistan; sajid.soofi@aku.edu (S.B.S.);; 3Sydney School of Public Health, Faculty of Medicine and Health, The University of Sydney, Sydney, NSW 2006, Australianeeloy.alam@sydney.edu.au (A.A.); michael.dibley@sydney.edu.au (M.D.)

**Keywords:** lactoferrin, preterm newborn, low birth weight, sepsis

## Abstract

**Background**: Sepsis remains a significant cause of morbidity and mortality in preterm and low birth weight (LBW) neonates, especially in low- and middle-income countries (LMICs). Lactoferrin, a glycoprotein present in breast milk with antimicrobial activity, is a low-cost, readily available, and promising intervention currently under investigation. The available literature presents conflicting results on the impact of lactoferrin on the risk of late-onset sepsis (LOS). This study evaluated the effectiveness of two doses of bovine lactoferrin (bLF) supplementation in preventing LOS and necrotizing enterocolitis (NEC) in preterm and LBW neonates in Pakistan. **Methods**: A three-arm, double-blind, placebo-controlled, randomized clinical trial in the neonatal intensive care unit of Aga Khan University was conducted from July 2019 to August 2020. Preterm (28 to 36 + 5 weeks gestational age) and low birth weight (≥1000 g to <2500 g) neonates who established enteral feeding by 72 h were eligible. The exclusion criteria included sepsis before randomization, maternal history of chorioamnionitis or group B streptococcus colonization, and congenital anomalies. Enrolled neonates were randomly assigned in a 1:1:1 ratio using a computer-generated random allocation sequence to receive placebo (D-glucose), 150 mg bLF, or 300 mg bLF mixed with breast milk once daily for 28 days. The study staff, parents, and outcome assessors were blinded to the allocation. The primary outcome was late-onset sepsis from the trial entry to 28 days. The secondary outcome was NEC from the trial entry to 28 days. Neonates were followed weekly for 28 ± 2 days, and episodes of LOS and NEC were recorded. **Results**: Of 305 neonates enrolled, 102, 102, and 101, respectively, were randomized to receive a placebo (arm A), 150 mg bLF (arm B), and 300 mg bLF (arm C), respectively. Outcome data of 291 participants (99 in arm A, 95 in arm B, and 97 in arm C) were available for inclusion in the intention-to-treat analysis. The frequency of culture-proven sepsis was 8/102 (7.8%) in arm A compared to 1/102 (0.98%) (*p* = 0.020) in arm B and 5/101 (4.9%) in arm C (*p* = 0.390). We did not find any difference in episodes of NEC between arms A (*n* = 3, 3%) and B (*n* = 0, 0%) (*p* = 0.087) or between arms A and C (*n* = 2, 2%) (*p* = 0.650). We reported compliance rates of 79 (79.79%) in arm A, 78 (82.1%) in arm B, and 82 (84.53%) in arm C for investigational products. Arm C recorded two deaths, but neither was attributed to the intervention. **Conclusions**: Bovine lactoferrin supplementation did not prevent late-onset sepsis in neonates of preterm and low birth weight in our trial. However, given the small sample size, further trials with larger sample sizes are required to investigate its efficacy in these at-risk groups.

## 1. Introduction

Approximately 1.3 million cases of neonatal sepsis are reported annually worldwide [1]. The neonatal immune system, which is inexperienced in adapting to the external environment, exhibits inadequate physical and chemical barriers, delayed T-cell response, and limited secretion of immunoglobulins. Consequently, newborns are highly susceptible to sepsis [2,3,4]. In preterm and low-birthweight (LBW) neonates, the risk of infection is further exacerbated by extended hospital stays and invasive procedures. Moreover, the likelihood of sepsis increases with decreasing gestational age and birth weight [5]. Neonatal sepsis has a case fatality rate of 2%, which escalates to 20% in preterm and LBW neonates [6] and is correlated with various forms of developmental delay among survivors [7].

As is extensively established, breastfeeding is vital in safeguarding preterm neonates against infections [8]. Lactoferrin, a whey protein in colostrum and mature breast milk, is hypothesized to play a pivotal role in the immune response [9]. Both in vivo and in vitro studies have demonstrated that lactoferrin exhibits antimicrobial and anti-inflammatory properties, modulates innate and adaptive immune responses, influences cytokine production, and affects the growth and migration of immune cells [10]. These attributes may explain the inverse relationship between sepsis, necrotizing enterocolitis (NEC), infant mortality rates, and the quantity of lactoferrin consumed through breast milk [11].

Bovine lactoferrin (bLF), like human lactoferrin, has been studied for its potential to prevent neonatal late-onset sepsis (LOS). However, conflicting results have been reported in randomized clinical trials, including the Lactoferrin Infant Feeding Trial (LIFT) [12] and Enteral Lactoferrin Supplementation for Preterm Infants Trial (ELFIN). While the meta-analysis of the LIFT trial suggested a significant reduction in LOS risk with bLF supplementation, the overall certainty of evidence remains low [13].

Most trials on bLF have been conducted in high-income countries, despite the potential benefits being particularly relevant for low- and middle-income countries (LMICs) with a high burden of neonatal sepsis [14]. Additionally, there was significant variation in the dosage of bLF used in previous trials, ranging from 100 mg to 300 mg, indicating a lack of consensus on the optimal dosage of bLF. We conducted a trial in Pakistan to address these gaps by using two different doses of bLF.

## 2. Methods

### 2.1. Trial Design, Setting, and Participants

This was a three-arm, double-blind, randomized controlled trial conducted in the NICU at the Aga Khan University Hospital (AKUH) in Karachi, Pakistan, in collaboration with the University of Sydney from July 2019 to August 2020. All preterm (28 to 36 + 5 weeks gestational age) and low birth weight (≥1000 g to <2500 g) neonates born at AKUH, with no signs of sepsis and tolerating oral feeds within 72 h of birth, were included in the trial. The exclusion criteria were the presence of a congenital anomaly, sepsis before enrollment, maternal history of chorioamnionitis or group B streptococcus colonization, and consent refusal. The protocol for this study has been previously published [15].

### 2.2. Randomization and Masking

Neonates who fulfilled the inclusion criteria were randomly assigned to receive either a placebo, D-glucose (arm A), 150 mg bLF (arm B), or 300 mg bLF (arm C) orally for 28 days. An independent statistician generated the random allocation sequence using STATA software (version 17), following a 1:1:1 ratio with equal probabilities for each treatment arm. A block size of nine was used for randomization to accommodate the expected number of participants recruited daily. In the case of multiple births, each neonate was randomized individually. Randomization codes were assigned to participant IDs and used to package the placebo and different doses of bLF, ensuring that the trial investigators, staff, and parents remained unaware of the sachet contents. Sachets that were indistinguishable in physical appearance were used for blinding purposes. Upon dissolution in milk, both the intervention and placebo formulations were matched in appearance, consistency, sensory characteristics, and color, ensuring concealment of group allocation throughout the study. The investigational product was sourced from Hilmar Ingredients in LA, USA, and packaged by Pharmaceutical Packaging Professionals (PPP) Ltd. in Melbourne, Australia.

### 2.3. Procedures

The investigational product (bLF or placebo) was dissolved in breast milk (minimum 5 mL) or formula milk when breast milk was unavailable. The first dose of the investigational product was administered within 72 h of birth by a clinical nurse in the NICU/postnatal ward/nursery of AKUH with the mother or primary caregiver as an observer. The nurse provided instructions and a pictorial brochure to the caregiver, demonstrating the investigational product preparation and the feeding method. Ward nurses supervised subsequent trial feeding throughout the neonate’s hospitalization. Upon discharge, caregivers received a one-week supply of the investigational product and were instructed to return unused empty sachets during weekly visits.

### 2.4. Primary Outcome

The trial’s outcome of interest was the incidence of late-onset sepsis (LOS), encompassing both clinical and culture-proven sepsis, from the time of trial entry to 28 days of life. Clinical sepsis is characterized by signs or symptoms, such as lethargy, apnea, respiratory distress, reluctance to feed, temperature instability, difficulty tolerating feeds, vomiting, aspiration, abdominal distention, and seizures. In addition, it required the presence of at least one abnormal laboratory parameter: (1) raised C-reactive protein level (>1 mg/dL), (2) deranged white blood cell count (<7.0 × 10^3^/uL or >23.0 × 10^3^/uL), (3) hypoglycemia (<40 mg/dL). A physician’s assessment of sepsis or administration of antibiotics by a clinician also qualifies as criteria for clinical sepsis. Culture-proven sepsis was defined as the presence of the above clinical signs and positive blood culture results.

### 2.5. Secondary Outcomes

The secondary outcomes were necrotizing enterocolitis diagnosed based on Bell’s criteria (Bell’s stages II and III) [16] from trial entry to the 28th day of life and neonatal mortality. We also compared the two doses of bLF to determine the optimal daily dose of bLF to prevent late-onset sepsis.

### 2.6. Data Collection

Data collection included daily information on the neonate’s clinical course, demographics, and maternal history during hospitalization. Weekly household follow-up visits were conducted until 28 ± 2 days of age to assess newborn well-being, feeding practices, feeding intolerance, history of illness, and treatment sought. Physical examination was performed for each visit. The first visit occurred at the Clinical Trial Unit of AKUH, followed by subsequent home visits. Body measurements were recorded at birth and again on the 14th and 28th days. To guarantee compliance, sachet retrieval and phone calls were conducted twice per week. The number of sachets was used to determine the adherence levels.

### 2.7. Sample Size

The sample size was calculated by assuming the incidence of neonatal sepsis in the placebo group to be 25% (based on the current hospital data) and the incidence of neonatal sepsis in the bLf group to be 8%, at 80% power with a 95% confidence level. The sample size was further adjusted to account for an estimated 10% loss to follow-up. Based on the above assumptions, the estimated sample size in each study arm was 95 (rounded off to 100) LBW neonates, resulting in 300 LBW neonates in the study.

### 2.8. Statistical Analysis

Intention-to-treat analysis was conducted using STATA version 17 (StataCorp, Stata Statistical Software, StataCorp LLC, College Station, TX, USA, 2019). Categorical data were reported as frequencies and percentages. For continuous variables, means and standard deviations (SD) or medians and interquartile ranges (IQR) were reported as appropriate. Chi-square or Fisher’s exact tests were used to compare categorical data between the placebo and each intervention arm as appropriate. The Wilcoxon rank-sum test was used to compare the anthropometric measurements among the groups.

### 2.9. Ethical Considerations

Approval for the study was obtained from the Ethical Review Committee (ERC) of Aga Khan University, Pakistan (ERC: 2020-0238-139190), University of Sydney, Australia (ID: 2017/420), and National Bioethics Committee (NBC), Pakistan (NBC-259). This trial was registered at ClinicalTrials.gov (NCT03431558). Written informed consent was obtained from the parents or legal guardians of the enrolled neonates after they read the patient information sheet and consent form. All serious adverse events (SAEs) and suspected unexpected serious adverse reactions (SUSARs) were reported within 24 h by ethics committees.

## 3. Results

Of the 532 neonates screened, 305 were enrolled in the study. Of these, 102 were allocated to the placebo arm, 102 to the arm receiving 150 mg bLF, and 101 to the arm receiving 300 mg bLF. On the 28th day of life, follow-up data were obtained from 291 participants. In comparison, 14 participants withdrew from the study for reasons such as migration, withdrawal of consent, or death (Figure 1).

Table 1 presents the baseline characteristics of the participants. Of the enrolled neonates, 85.2% (*n* = 260) were delivered via cesarean section, while 14.8% (*n* = 45) were born through spontaneous vaginal delivery. The majority of patients (54.3%; *n* = 165) were female. The mean gestational age and birth weight were 34 weeks and 1900 g, respectively. Surfactant was administered to 4.6% (*n* = 14), while 29.5% (*n* = 90) received antibiotics for early onset sepsis. Among the neonates, 52.8% (*n* = 161) required respiratory support, with 95.6% (*n* = 154) receiving noninvasive ventilation (62.1% CPAP and 33.5% high-flow oxygen), and 4.3% (*n* = 7) received invasive ventilation. During the trial period, approximately 51.5% (*n* = 150) of the neonates were exclusively breastfed, 48.1% (*n* = 140) received mixed feeding, and only 0.3% (*n* = 1) were exclusively formula fed.

A sub-analysis was conducted to evaluate differences in the frequency of culture-proven sepsis among the three study arms. The highest number of cases was observed in the placebo arm (*n* = 8), followed by the 300 mg arm (*n* = 5), and the lowest in the 150 mg arm (*n* = 1). There was a statistically significant difference in the number of episodes of culture-confirmed late-onset sepsis (LOS) between the placebo arm (eight cases, 8%) and the arm receiving 150 mg bLF (one case, 1%), with a *p*-value of 0.02. However, the incidence of culture-proven LOS was similar between newborns administered 300 mg bLF (6 cases, 6%) and those in the placebo arm (*p* = 0.39). No significant differences were observed in either culture-proven or clinical sepsis between the placebo and 150 mg bLF groups (*p* = 0.140) or between the placebo and 300 mg bLF groups (*p* = 0.470) (Table 2).

A total of seven cases were diagnosed as Pre-NEC or Stage I or II NEC and managed conservatively with antibiotics. Among these, three had positive blood cultures, and two of them had co-infections. There were four cases diagnosed as Stage III NEC in the placebo arm, and all but one of these cases had culture-proven sepsis. The specific organisms isolated are detailed in Table 3.

However, no statistically significant differences were found in the occurrence of necrotizing enterocolitis (NEC) between the placebo and 150 mg bLF groups (*n* = 3, 3% vs. *n* = 0, *p* = 0.09) or between the placebo and 300 mg bLF groups (*n* = 3, 3% vs. *n* = 2, 2%, *p* = 0.65).

Altogether, 38 newborns required hospitalization after discharge due to sepsis or necrotizing enterocolitis (NEC) as follows: 13.1% (13 newborns) from the placebo arm, 9.5% (9 newborns) from the 150 mg bLF arm, and 16% (16 newborns) from the 300 mg bLF arm. The median length of hospital stay for sepsis or NEC was the highest in the placebo arm: 14 days (IQR: 4–22), 7 days (IQR: 4–12) in the 150 mg bLF arm, and 6 days (IQR: 3–10.5) in the 300 mg bLF arm.

Anthropometric measurements, including weight, head circumference, and length, were recorded for the participating newborns and were tracked until day 28. No significant differences were detected in weight or average weight gain between the three groups on both the 14th and 28th days (Table 4). We also did not document any difference in head circumference and length amongst the three groups.

Two fatalities occurred in the arm that received 300 mg bLF during the trial. However, it was concluded that these incidents were not connected to the trial intervention.

Among the 291 neonates who completed the study, 79 (79.79%) in the placebo arm, 78 (82.1%) in the 150 mg bLF arm, and 82 (84.53%) in the 300 mg bLF arm consumed the investigational product (IP). Compliance, assessed by the number of sachets used per 30 days, was comparable between the three arms (Table 5).

## 4. Discussion

In our study, the number of culture-proven late-onset sepsis (LOS) episodes in preterm and low-birthweight (LBW) neonates was lower in Arm A, which received enteral supplementation with 150 mg of bovine lactoferrin (bLF), compared to the placebo group; however, the difference was not statistically significant. Similarly, a higher dose of bLF was associated with fewer LOS episodes than both the 150 mg bLF and placebo groups, but this difference was also not statistically significant.

Previous trials on enteral supplementation with bLF have demonstrated variable effectiveness of the intervention on LOS, reporting risk ratios ranging from 0.24 to 1.05 [17]. The variances between the trials can be attributed to the different care practices in the facilities where the trials were conducted and the different doses, regimes, and sources of lactoferrin used in the studies. Our study used specific dosing; however, others have used dosing of LF as mg per kg [18].

In addition to the current study, we found only one other trial of lactoferrin conducted in an LMIC [19]. Neonatal sepsis is estimated to be 2–3 times higher in LMICs than in high-income countries [20]. Poor sanitation, low socioeconomic status, maternal malnutrition, overcrowding, low exclusive breastfeeding rates, higher prematurity and low birthweight rates, and suboptimal quality of care significantly increase the risk of neonatal sepsis in low-resource settings [21,22,23,24]. Moreover, difficulties in the early diagnosis of sepsis and the use of broad-spectrum antimicrobials with the subsequent emergence of multidrug-resistant organisms increase morbidity and mortality in low-resource settings [25,26,27]. This study was conducted in a private tertiary care hospital with a well-established sepsis-screening protocol in an urban setting. Therefore, it is possible that many of the participants included in the trial did not have most of the risk factors unique to a resource-poor population. Kaur et al., who conducted their trial in India with a sociodemographic profile similar to Pakistan, showed lactoferrin’s protective effect on neonatal sepsis [19]. Additionally, Ochoa et al. demonstrated that infants with lower breastmilk intake are more likely to benefit from bLF supplementation [28]. These findings merit further studies in LMICs, as the potential preventative benefits of lactoferrin are likely to have the greatest impact in settings where breastfeeding rates are low and the incidence of sepsis is high. Lack of access to or poor quality of care prevents appropriate and timely management.

Although we did not observe a difference in the incidence of NEC across the study arms, research using animal models has shown that lactoferrin could potentially protect against necrotizing enterocolitis (NEC). However, human studies have not corroborated these findings [29]. A meta-analysis of randomized controlled trials reported a risk ratio of 1.10 (95% CI, 0.86 to 1.41) for developing NEC among neonates receiving lactoferrin compared to placebo [17].

Animal studies also suggested that lactoferrin may improve nutrient absorption and growth [30]. However, similar to previous human studies [31], our trial did not record any effect of lactoferrin supplementation on the growth rates of preterm infants.

Our study has notable strengths, including a high compliance rate (97%). Compliance was ensured through reminders and home visits. Before the trial, a TIPS was conducted to ensure acceptability and develop clear usage instructions.

This study has some limitations. The incidence of neonatal sepsis was the outcome of interest. Importantly, our trial included a relatively small group of 300 infants. A larger study group might have resulted in more sepsis events, which would offer a more thorough understanding of the implications of lactoferrin supplementation. Although, understandably, families choose physicians close to their homes for convenience after hospital discharge, this may have led to some episodes of sepsis being missed in our study.

## 5. Conclusions

Enteral supplementation with bLF did not demonstrate a significant effect on late-onset sepsis in preterm and low-birthweight infants in our cohort. However, more extensive studies are needed, especially in resource-limited low- and middle-income countries (LMICs). Future studies should also investigate the optimal lactoferrin dosages.

## Figures and Tables

**Figure 1 nutrients-17-01774-f001:**
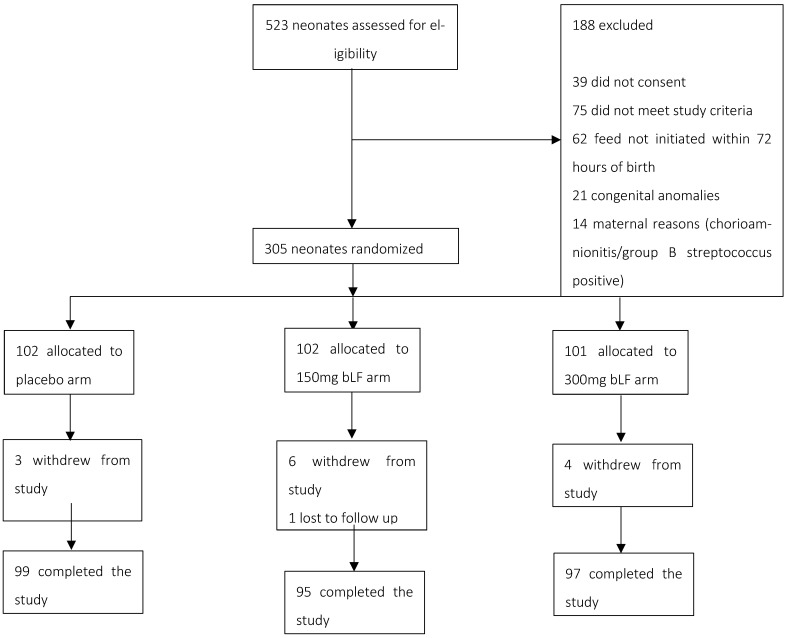
Trial profile.

**Table 1 nutrients-17-01774-t001:** Baseline characteristics of enrolled neonates.

Characteristics	Placebo	150 mg bLF	300 mg bLF
	*n* = 102	*n* = 102	*n* = 101
Gestational age (weeks)	34.0 (32.0–35.0)	35.0 (33.0–36.0)	34.0 (32.0–35.3)
Birth weight (grams)	1885.0 (1520.0–2100.0)	1960.0 (1700.0–2100.0)	1900.0 (1600.0–2100.0)
Length (cms)	43.5 (41.1–45.2)	44.0 (42.3–45.3)	43.8 (41.2–45.5)
Head circumference (cms)	30.8 (29.5–32.0)	31.1 (29.8–32.3)	31.0 (29.6–32.0)
Sex of the child			
Male	46 (45.1%)	44 (43.1%)	49 (48.5%)
Antibiotics received	32 (31.4%)	28 (27.5%)	30 (29.7%)
Mode of delivery			
Spontaneous vaginal delivery	17 (16.7%)	18 (17.6%)	10 (9.9%)
Caesarean section	85 (83.3%)	84 (82.4%)	91 (90.1%)
Length of stay (days)	5.5 (3.0–8.0)	3.0 (2.0–6.0)	4.0 (2.0–8.0)
Surfactant received	3 (2.9%)	5 (4.9%)	6 (5.9%)
Oxygen received	61 (59.8%)	48 (47.1%)	52 (51.5%)
Type of oxygen			
CPAP	19 (31.1%)	15 (31.3%)	20 (38.5%)
Mechanical ventilator	2 (3.3%)	3 (6.3%)	2 (3.8%)
Nasal prongs	40 (65.6%)	30 (62.5%)	30 (57.7%)
Age at recruitment (days)	2.0 (2.0–3.0)	2.0 (2.0–3.0)	2.0 (2.0–3.0)
Feeding method			
Exclusive breastfeeding	48/99 (48.5%)	46/95 (48.4%)	56/97 (57.7%)
Mixed feeding	50/99 (50.5%)	49/95 (51.6%)	41/97 (42.3%)
Formula milk feeding	1/99 (1.0%)	0/95 (0.0%)	0/97 (0.0%)

Data presented as median (IQR) and *n* (%).

**Table 2 nutrients-17-01774-t002:** Primary and secondary outcomes.

Outcome Measure	Arm			Placebo vs. 150 mg bLF *p*-Value	Placebo vs. 300 mg bLF *p*-Value
	Placebo *n* = 99	150 mg bLF *n* = 95	300 mg bLF *n* = 99		
Culture-proven sepsis, *n* (%)	8 (8%)	1 (1%)	5 (5%)	0.020	0.390
Presumed sepsis, *n* (%)	3 (3%)	4 (4%)	3 (3%)	0.660	>0.999
Presumed and culture-proven sepsis, *n* (%)	11 (11%)	5 (5%)	8 (8%)	0.140	0.470
NEC, *n* (%)	3 (3%)	0 (0%)	2 (2%)	0.087	0.650
* Pre-NEC, *n* (%)	4 (4%)	0 (0%)	3 (3%)	0.048	0.700
Mortality	0	0	2	-	-

* Pre-NEC was defined as the presence of any one or more than one sign/symptom: lethargy, feeding intolerance (abdominal distension, vomiting, or aspirates), blood in stools, apnea prompting cessation of feeds, gastric decompression, and initiation of antibiotics following septic workup. It was labelled as NEC Stage I or II.

**Table 3 nutrients-17-01774-t003:** Microorganisms isolated in culture-proven sepsis across the three groups.

Fungal	Gram-Negative	Gram-Positive
Aspergillus Fumigatus	Bacilli	Corynebacterium species
	Citrobacter Freundi	Micrococcus Species
	Escherichia Coli	Staphylococcus Aureus
	Klebsiella Pneumonia	Staphylococcus Species

**Table 4 nutrients-17-01774-t004:** Weight gain by treatment arm.

	Placebo		150 mg bLF		300 mg bLF		Placebo vs. 150 mg bLF *p*-Value	Placebo vs. 300 mg bLF *p*-Value
	Median (IQR)	*n*	Median (IQR)	*n*	Median (IQR)	*n*		
Weight (gm) ᵠ								
At enrollment	1885 (1520.0–2100.0)	102	1960 (1700.0–2100.0)	102	1900 (1600.0–2100.0)	101	0.120	0.880
14th day of life	2155 (1750.0–2480.0)	96	2200 (1985.0–2490.0)	96	2032.5 (1780.0–2440.0)	98	0.270	0.680
28th day of life	2537.5 (2060.0–2900.0)	94	2515 (2220.0–2900.0)	94	2405 (2070.0–2800.0)	96	0.380	0.550
Average weight gain (gm) ᵠ								
Enrollment on the 28th day of life	655.0 (420.0–830.0)	94	620.0 (450.0–805.0)	94	580.0 (420.0–770.0)	96	0.800	0.220

ᵠ Compared using Wilcoxon rank-sum.

**Table 5 nutrients-17-01774-t005:** Compliance and hospitalization recorded for all treatment arms.

	Arm			Placebo vs. 150 mg bLF *p*-Value	Placebo vs. 300 mg bLF *p*-Value
	Placebo *n* = 99	150 mg bLF *n* = 95	300 mg bLF *n* = 97		
Children who took all doses, *n* (%)	79 (79.8)	78 (82.1)	82 (84.5)		
Compliance (sachets used/30 days) *	95.7%	98.3%	98.6%	0.093	0.055
Children hospitalized, *n* (%)	13 (13.1)	9 (9.5)	16 (16.5)		
Length of stay in days ᵠ, median (IQR)	14 (4.0–22.0)	7 (4.0–12.0)	6 (3.0–10.5)	0.260	0.120

* Compared using a two-sample *t*-test. ᵠ Compared using Wilcoxon rank-sum.

## Data Availability

The data presented in this study are available on request from the corresponding author due to ethical reasons.

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
