# Peer review of "Evaluation of Bovine Lactoferrin for Prevention of Late-Onset Sepsis in Low-Birth-Weight Infants: A Double-Blind Randomized Controlled Trial"

_nutrients, 2025, doi:10.3390/nu17111774_

Round 1
Reviewer 1 Report
Comments and Suggestions for Authors
Excellent study and well written.
- Abstract: there are 4 numbers for patient groups. It appears like an extra "102"
- Since mortality data is only in the abstract, add that no patients died in groups 1 and 2 or add data for all 3 groups for outcomes Table 2.
- Include in methods how blinding was done specifically
- 2.4 and 2.5 both are labeled outcomes and have overlap. Suggest to make 2.4 Primary outcome and 2.5 Secondary Outcomes.
- Trial profile: instead of "outside study area"-maybe "did not meet study criteria" or please be more specific
- Profile, why did patients withdraw?
- In Results, first paragraph after Table 1, is the antibiotics refer to early onset sepsis antibiotics? Please add that clarification
- Is there other infection related data such as urinary tract infections? if so, would be good to include.
- Can the cause of infections (gram positive, gram negative, fungal) and specific pathogens (e.g, Staph aureus, CoNs, Klebsiella, Candida etc..) be in a table or text?
- Any data to share of Surgical (Nec Stage III) and Medical NEC Stage II?
- Please define pre-NEC in table 2?
- Is there data to share for Antibiotic days between groups after enrollment?
- Are other outcomes available to report related to growth: head circumference between groups?
- I recognize numbers are small. Would a sentence in the results or discussion of sub-analysis for Placebo vs both lactoferrin groups for (1) culture proven sepsis, (2) NEC and (3) culture proven sepsis or NEC be feasible?
- Any differences if subanalysis for younger infants: 1000-1500 grams or 28-32 weeks available to include?
- In discussion (and/or introduction) would also include that some studies used specific dosing as in this study vs dosing of LF as mg per kg.
excellent
Author Response
Comment 1: Abstract: there are 4 numbers for patient groups. It appears like an extra "102"
Response: Thank you for pointing out the error. It is corrected
Comment 2: Since mortality data is only in the abstract, add that no patients died in groups 1 and 2 or add data for all 3 groups for outcomes Table 2.
Response: The data is added in Table 2
Comment 3: Include in methods how blinding was done specifically
Response: Sachets that were indistinguishable in physical appearance were used for blinding purposes. Upon dissolution in milk, both the intervention and placebo formulations were matched in appearance, consistency sensory characteristics and color, ensuring concealment of group allocation throughout the study. (details added)
Comment 4: 2.4 and 2.5 both are labeled outcomes and have overlap. Suggest to make 2.4 Primary outcome and 2.5 Secondary Outcomes.
Response: We have revised paragraph 2.4 as Primary Outcome and 2.5 as secondary outcomes
Comment 5: Trial profile: instead of "outside study area"-maybe "did not meet study criteria" or please be more specific
Response: Done replaced outside study area" with did not meet study criteria"
Comment 6: Profile, why did patients withdraw?
Response: There were migrations within the city and outside the city, we were unable to locate their new address, withdrawal of consent by 5 participants as the mother-in law- and husband did not agree to administer the sachets and follow up in clinics. This is already mentioned in the Result section of the 1st Paragraph, highlighted for clarity.
Comment 7: In Results, first paragraph after Table 1, is the antibiotics refer to early onset sepsis antibiotics? Please add that clarification
Response: Added ,it refers to early onset sepsis
Comment 8: Is there other infection related data such as urinary tract infections? if so, would it be good to include.
Response: In patients that underwent triple tap, UTI was documented in only 1 culture .
Comment 9: Can the cause of infections (gram positive, gram negative, fungal) and specific pathogens (e.g, Staph aureus, CoNs, Klebsiella, Candida etc..) be in a table or text?
Response: Yes, we have added Table 3 and have changed the serial numbers accordingly highlighted
Comment 10: Any data to share of Surgical (Nec Stage III) and Medical NEC Stage II?
Response: Data added and highlighted in the text in the result section
Comment 11: Please define pre-NEC in table 2?
Response: Pre-NEC was defined as the presence of any one or more than one sign and symptoms such as lethargy, feeding intolerance (abdominal distension, vomiting, or aspirates), blood in stools, apnea, prompting cessation of feeds, gastric decompression, and initiation of antibiotics following septic workup. These were later defined as either Stage 1 or Stage 2 NEC based on Bells criteria. Definition added in table 2.
Comment 12: Is there data to share for Antibiotic days between groups after enrollment?
Response: Unfortunately, we do not have the data for antibiotic days
Comment 13: Are other outcomes available to report related to growth: head circumference between groups?
Response: Yes, we have them available, added in 5th paragraph and highlighted. No difference was documented in Head circumference or length.
Comment 14: I recognize numbers are small. Would a sentence in the results or discussion of sub-analysis for Placebo vs both lactoferrin groups for (1) culture proven sepsis, (2) NEC and (3) culture proven sepsis or NEC be feasible?
Response: We have mentioned this in the results section and highlighted
Comment 15: Any differences if subanalysis for younger infants: 1000-1500 grams or 28-32 weeks available to include?
Response: Numbers are very small for sub-analysis
Comment 16: In discussion (and/or introduction) would also include that some studies used specific dosing as in this study vs dosing of LF as mg per kg.
Response: Thank you for this helpful suggestion. We have now included the information in the revised version of the manuscript.

Reviewer 2 Report
Comments and Suggestions for Authors
I have little criticism for this well-conducted trial and its well-written manuscript, except for the statistical handling of the data and the ensuing conclusions. Analysis of a three-arm RCT with low numbers of events requires a 2x3 Fisher exact test for the primary endpoint. With a p-value of 0.0596292, the trial has clearly a negative finding (lactoferrin does not work), and only this should be given in the conclusions (Abstract, fist paragraph of discussion, last paragraph of dicussion). Post-hoc analyses after a negative primary result are hypothesis-generating at best and entail a high risk to yield spurious results, more so if no p-value correction is made for multiple comparisons.
Author Response
Comment: I have little criticism for this well-conducted trial and its well-written manuscript, except for the statistical handling of the data and the ensuing conclusions. Analysis of a three-arm RCT with low numbers of events requires a 2x3 Fisher exact test for the primary endpoint. With a p-value of 0.0596292, the trial has clearly a negative finding (lactoferrin does not work), and only this should be given in the conclusions (Abstract, first paragraph of discussion, last paragraph of discussion). Post-hoc analyses after a negative primary result are hypothesis-generating at best and entail a high risk to yield spurious results, more so if no p-value correction is made for multiple comparisons.
Response: Thank you for your thoughtful and detailed feedback. We acknowledge the concern regarding the statistical handling of the primary endpoint. As suggested, we have revised the abstract, discussion, and conclusion to reflect the suggested changes.
Round 2
Reviewer 2 Report
Comments and Suggestions for Authors
o.k., no further comments